# Deposition of Self-Lubricating Coatings via Supersonic Laser Deposition (SLD)

**Nicholas Soane, Andrew Cockburn *** , **Martin Sparkes and William O'Neill**

Institute for Manufacturing (IfM), University of Cambridge, 17 Charles Babbage Road, Cambridge CB3 0FS, UK; nvsoane@gmail.com (N.S.); mrs46@cam.ac.uk (M.S.); won207@cam.ac.uk (W.O.)
* Correspondence: ac282@cam.ac.uk; Tel.: +44-(0)1223-748265

**Abstract:** This paper describes the use of Supersonic Laser Deposition (SLD) to manufacture nickel/graphite composite coatings on titanium and aluminium substrates. Laser heating is critical for depositing coatings containing up to 13.3 vol% graphite. For a given feedstock composition, the resulting graphite content and average size of the graphite particles retained in the coating increases with laser power, until substrate melting occurs. The effect of the powder type, feedstock composition, and process conditions on coating structure is characterized. The friction and wear behaviour of selected coating compositions is examined. Nickel coatings containing 13.3 vol% graphite demonstrated self-lubricating behaviour with a stable coefficient of friction below 0.14 in pin-on-disc testing.

**Keywords:** Supersonic Laser Deposition; solid lubricant; metallic coating; nickel-coated graphite; pin-on-disc testing





## 1. Introduction

Friction and wear of mechanical components pose a major engineering challenge. Liquid lubricants perform well in most cases, but they are unsuitable for applications where loads and temperatures are high, or atmospheric pressure is low.

Since the 1960s, researchers have worked to develop metal-based materials with embedded solid lubricants, such as graphite, molybdenum disulphide (MoS$_2$), and hexagonal boron nitride (hBN), that can withstand harsh conditions [1]. Early work showed that composite materials comprising metals and solid lubricants produced via powder metallurgy (sintering at high temperatures and mechanical compaction) could be self-lubricating under extreme conditions [2]. The metal matrix provides structure whereas the solid lubricant embedded throughout the composite reduces friction. The proportion of lubricant necessary is dependent on the lubricant used. Typical values of graphite content vary between 6 and 20 vol% for iron-, aluminium- [1], nickel- [3] and copper-based [4] composites. When using hBN, lubricant fractions up to 10% have been reported as optimal [5].

Sintered metal and solid lubricant composites are manufactured as discrete blocks or separate components, which must then be bonded or assembled in place. Depositing coatings directly would simplify application [6], and has been the subject of research using processes such as plasma spray [7,8], high velocity oxy-fuel spraying [9], and laser cladding [10]. For all the above techniques, the effect of the solid lubricants is limited, as they are degraded while being deposited.

The degradation of lubricants has led to the exploration of process and feedstock modifications in order to allow solid lubricants to be retained. Zhao et al. [11] utilized nano scale copper particles to enable hBN to be incorporated into coatings. Although the volume fraction of hBN added was relatively low (1.25 wt%), TEM analysis confirmed hBN was present in the deposited coatings, and a 20–30% reduction in the coefficient of friction (CoF) was observed. Yan et al. [12], explored the laser cladding of Ni/hBN self-lubricating composites containing 5 and 10 wt% hBN. X-ray analysis suggested that little to no hBN

remained in coatings deposited from feedstocks containing uncoated hBN. When the hBN was ball milled with nano scale nickel powder prior to cladding to provide a thin nickel coating to the hBN particles, hBN was retained in the coating and resulted in a reduced friction coefficient of 0.36 in a pin-on-ring test.

A suspension containing hBN was introduced into a modified high velocity air fuel thermal spray system by Ganvir et al. [13] alongside $Cr_3C_2$–NiCr powder, avoiding the need for pre-deposition mixing and milling of the feedstock. This method resulted in coatings containing hBN, which were then compared with hBN-free coatings deposited under similar conditions. Despite having lower hardness and increased porosity, the composite coatings containing hBN demonstrated reductions in both CoF and wear rate during pin-on-disc testing when compared with coatings deposited using only the $Cr_3C_2$–NiCr feedstock.

The lower operating temperatures in cold spray (CS) make the process a more attractive option for the deposition of self-lubricating coatings. However, the addition of uncoated lubricant particles has been found to reduce deposition efficiency significantly. Feedstocks containing hBN [14] and $MoS_2$ [15] have been sprayed using CS, but the addition of solid lubricant particles has been shown to inhibit deposition. Only feedstocks with lubricant fractions $\leq$ 5 vol% are reported to form coatings, whereas improvements in tribological performance have not been demonstrated.

Nikbakht and Jodoin [16] explored the deposition of Cu/hBN coatings using both CS and pulsed gas dynamic spray. While CS was unable to deposit Cu/hBN, pulsed gas dynamic spray was able to deposit coatings containing between 9 and 16 vol% hBN. The effect of the addition of hBN on friction and wear behaviour was not reported.

Feedstocks containing metal/lubricant composite particles are more effective. Huang et al. [17] prepared Al/nickel-coated graphite (NCG) coatings containing graphite fractions of up to 15% using CS. Coefficients of friction were unstable, rising from as low as 0.16 to 0.22 during ball on disc testing. Eden et al. [18,19] compensated for the fragility of the solid lubricants by encapsulating solid lubricants in a thick metal layer prior to spraying. Nickel-coated hBN powders were synthesized with a nickel fraction of up to 68 vol%. This allowed hBN to be incorporated into nickel coatings. The CoF for the coatings (0.2) was approximately half that reported for pure nickel.

The use of metal/lubricant clusters has also been demonstrated. Neshastehirz et al. [20] encapsulated μm scale hBN particles in nickel using electroless plating, resulting in feedstock powders containing 1 wt% hBN. Coatings deposited using cold spray showed CoFs of 0.35, a reduction of approximately 40% relative to CS nickel. Wang et al. [21] used mechanical alloying to prepare a feedstock consisting of 1 wt% boron nitride nanosheets encapsulated in a copper matrix. When deposited, the resulting coatings displayed lower CoF (0.5) and wear rate than either copper substrates or copper cold spray deposits.

Although CS has been used to deposit coatings containing solid lubricants, this has either required extensive process modification or the preparation of specific feedstocks. Neither approach has produced coatings which demonstrate stable CoF values below 0.2.

Supersonic Laser Deposition (SLD) is a hybrid process, which expands on the advantages of CS, high rate ($\leq$10 kg·hr$^{-1}$) deposition without melting, by locally heating the substrate at the deposition site. Laser heating softens the substrate, improving bonding and allowing particles to deposit at impact velocities below the critical velocity for CS. Lower impact velocities reduce operating costs by reducing the need for gas heating and allowing the use of helium as the process gas to be avoided [22]. The improved bonding arising from the softened impact site also allows high strength materials such as tungsten [23] and cobalt chrome alloys [24] to be deposited.

In SLD, the softened substrate and reduced particle velocity reduces powder deformation when compared with CS, whereas the temperature remains below that found in laser cladding or plasma spray. Reduced substrate hardness has been shown to increase the fraction of coated graphite particles which remain encapsulated on impact [17], reducing the fraction of free solid lubricant present at the impact site to inhibit deposition. This

suggests that SLD may be able to more readily deposit metal coated solid lubricant particles than cold spray alone.

The purpose of this paper is to explore the feasibility of using SLD to produce self-lubricating metallic coatings. Aluminium [25] and titanium [26] substrates were chosen, as they are light weight structural materials, which suffer from adhesive wear, high coefficients of friction, and poor wear resistance, making the development of low friction coatings for both materials attractive. Graphite was selected as the solid lubricant due to the availability of graphite powder in both nickel-coated and uncoated forms. Nickel was selected as the metallic matrix, as it is known to readily deposit using SLD, and it removed the chance of forming brittle intermetallic phases between the nickel-coated graphite shells and the matrix material.

In this paper, the deposition of feedstocks containing coated and uncoated graphite particles is compared. The CoF and wear rate of selected coatings are characterized using pin-on-disc testing.

## 2. Materials and Methods

### 2.1. SLD Equipment

A simplified schematic of the SLD system is shown in Figure 1. This uses a proprietary cold spray system, which operated with 30 bar, 500 °C nitrogen throughout the study. The substrate is heated by a 4 kW, 1070 nm wavelength fibre laser (IPG photonics GmbH, Burbach, Germany) operating beyond focus to provide a 6 mm diameter spot at the substrate. Deposition temperature is recorded via an Impac IGAR 12-LO infra-red pyrometer (Lumasense Inc., Santa Clara, CA, USA). The pyrometer allows the SLD system to operate under closed loop control. Temperature data can be fed into a proportional-integral-differential controller which modulates laser power to maintain a consistent deposition site temperature.

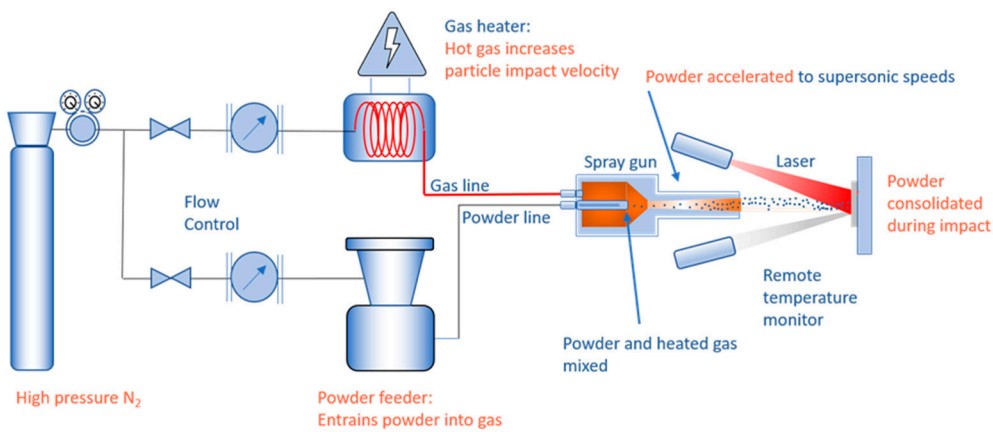

**Figure 1.** Schematic diagram showing SLD equipment.

### 2.2. Materials

For this experiment, 3 mm thick 6082-T6 aluminium and grade 2 titanium were chosen as substrates. Substrate preparation involved grinding with 320 grit SiC paper, followed by degreasing using ethanol.

The feedstock powders chosen for this study include spherical nickel powder to act as the matrix, and graphite flakes and nickel-coated graphite (NCG) as the solid lubricants. Details of the feedstock powders are provided in Table 1. Polished sections of the powders are shown in Figure 2.

Powder mixtures were shaken together in sealed containers for 5 min prior to being loaded into the powder feeder, in order to ensure consistent powder mixing. The feedstock was homogeneous and flowed evenly when loaded into the feeder. All powders were stored in a heated cabinet to prevent moisture condensation. Similarly, a heating blanket

strapped around the powder feeder was used to keep the feedstock powder between 50 °C and 55 °C once loaded in the feeder. Throughout this paper, all constituents in a powder mixture are reported by their volume fraction. An SEM image of a mixture of nickel and NCG feedstock powders helps illustrate the difference in particle size and shape. A mixture of two different powders, NCG (17% graphite by volume) and pure nickel, is shown in Figure 3.

**Table 1.** Details of feedstock powder.

| Powder | Synthetic Graphite | Nickel-Coated Graphite (NCG) | Nickel |
|---|---|---|---|
| Supplier | Inoxia | Oerlikon Metco | Sandvik Osprey |
| Product Name | Graphite Powder | Metco 308NS-1 | Ni (MIM) |
| Nominal Particle Sizes | 95% < 53 μm | 30–90 μm | D50 = 24.7 μm |
| External Morphology | Irregular, lamellar | Irregular Ni shell | Spherical |
| Internal Morphology | Lamellar, dense | Lamellar, dense | Solid, dense |
| Method of Manufacture | Petrochemical | Hydrometallurgy autoclave | Gas atomization |
| % Metal by Weight | 0% | 85% | 100% |
| % Lubricant by Weight | 100% | 15% | 0% |

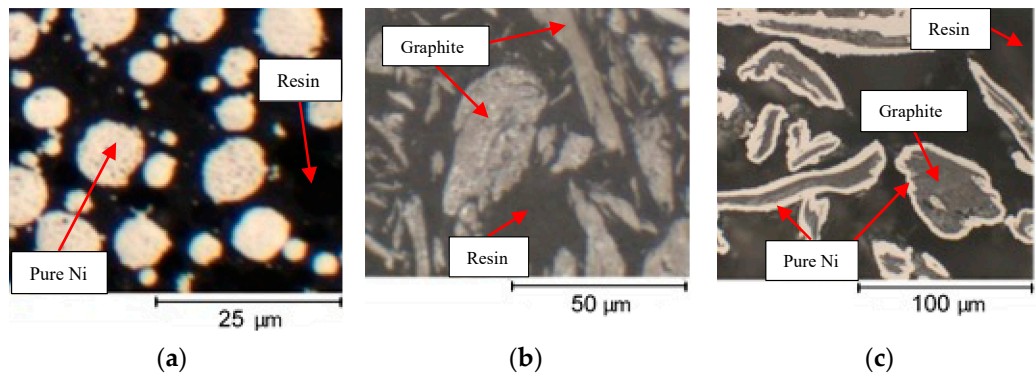

(**a**)                                        (**b**)                                        (**c**)

**Figure 2.** Micrographs showing polished cross sections of nickel (**a**), graphite (**b**), and NCG (**c**) powders.

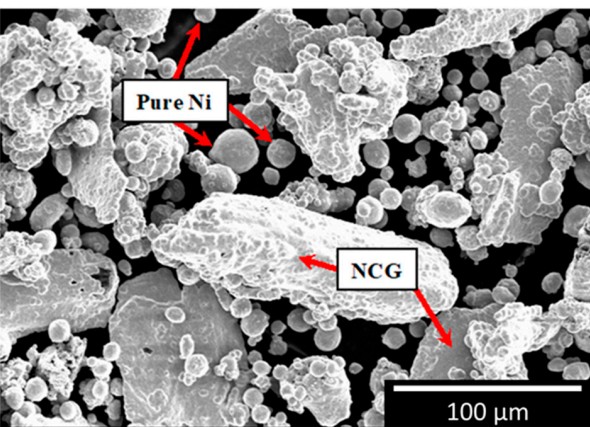

**Figure 3.** SEM image showing a mix of a pure nickel and NCG powder.

### 2.3. Deposition

A series of single lines were deposited using 30 bar, 500 °C nitrogen, and the parameters given in Table 2. Unlike laser power, which could be programmed to rapidly vary during each experiment, the powder mixtures were handmade. Thus, a select number of mixtures were tested, with pure NCG equaling 41 vol% graphite being the highest. Only two traverse rates were used, since its effect was not a central part of the investigation.

**Table 2.** Parameters for single line deposition trials.

| Substrate | Metal Powder | Lubricant Powder | Graphite (vol%) | Traverse Rate (mms$^{-1}$) | Laser Power (W) |
|---|---|---|---|---|---|
| Al 6082 | Ni | Graphite | 0, 17, 41 | 40, 60 | 0–1500 |
| Al 6082 | Ni | NCG | 0, 17, 41 | 40, 60 | 0–1500 |
| Ti Gd 2 | Ni | Graphite | 0, 17, 41 | 40, 60 | 0–1500 |
| Ti Gd 2 | Ni | NCG | 0, 17, 41 | 40, 60 | 0–1500 |

### 2.4. Characterisation

Samples were cut using a precision saw, Secotom-10, Struers; Copenhagen, Denmark, mounted using a hot press, Opal 400, ATA GmbH; Frankfurt, Germany, ground and polished with a metallographic polishing machine, Saphir 550, ATA GmbH; Frankfurt, Germany, to a final polish of 0.04 µm colloidal silica. Polished sections were examined using an Olympus BX51 optical microscope, Olympus Europa; Hamburg, Germany. A4i, Aquinto, Germany, and ImageJ, open-source image analysis packages were used to measure track dimensions, porosity, and graphite fraction.

The friction and wear tests were carried out by ESR Technology Ltd. in Warrington, UK, according to ASTM-G99. A 50 mm radius, 0.1 µm Ra EN31 steel pin, hardened and tempered to 66–68 Rockwell hardness C, was applied to rotating samples with a 20 N load.

Wear rate was determined through the measurement of the volume of material removed using white light interferometry.

The high friction of the pure nickel coatings exceeded the load rating of the tribometer. A reduction in rotations per minute (RPM) from 150 to 50 was required to enable the nickel coating to be tested. To compensate for the variable RPM, tests were run for different durations to maintain a similar total sliding distance for each sample. The specific wear testing regime and settings used are listed in Table 3.

**Table 3.** Load parameters for the wear tests.

| Graphite Content (vol%) | Number of tests | Load (N) | Diameter (mm) | RPM | Run Time (min) |
|---|---|---|---|---|---|
| 0 | 3 | 20 | 23 | 50 | 30–125 |
| 3.1 | 4 | 20 | 20, 23 | 150 | 45 |
| 13.3 | 3 | 20 | 18, 22, 23 | 100 | 68 |

## 3. Results

### 3.1. Deposition and Graphite Inclusion

Build rate increased with laser power for all powders tested, as shown in Figure 4. The pure nickel and the Ni/NCG feedstocks containing 17 vol% graphite both deposited using CS. The other powder mixtures required laser heating in order to allow deposit formation.

Micrographs of single lines deposited onto aluminium with 1500 W laser power using pure nickel, nickel/graphite, and nickel/NCG, depicted in Figure 5, show that although the nickel/graphite mixture has deposited, no graphite is visible. This contrasts with the nickel/NCG deposit, where graphite inclusions are observed.

An example of track profile development with respect to increasing input laser power is given in Figure 6, showing the transition from zero deposition to eventual substrate erosion.

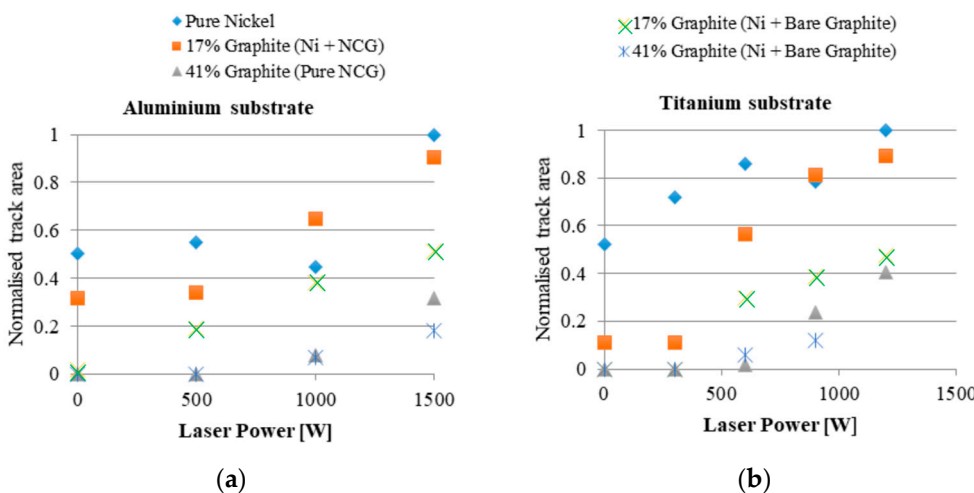

**Figure 4.** Graphs showing track cross sectional area vs. laser power for various feedstock compositions on both aluminium (**a**) and titanium (**b**) substrates at a traverse rate of 60 mms$^{-1}$.

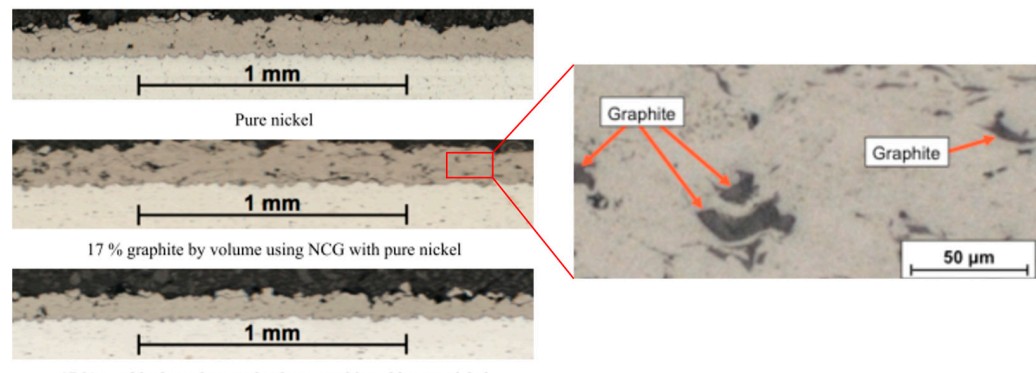

**Figure 5.** Micrographs showing single passes deposited from nickel, nickel + NCG, and nickel + graphite on aluminium using 1500 W. Graphite can be seen in the nickel + NCG coating.

Graphite content and porosity as a function of temperature for a 30 vol% graphite Ni + NCG feedstock is shown in Figure 7 for aluminium and titanium substrates. The maximum graphite content observed in the deposits is 17 vol%.

The average observed graphite particle size retained in deposited Ni/NCG tracks on both aluminium and titanium increased with input laser power, as shown in Figure 8. In cases where the graphite size is reported as zero, no coating deposition occurred.

Cross-sections of tracks deposited using feedstocks containing 30 and 41 vol% graphite are shown in Figures 9a and 9b, respectively. The graphite flakes visible in the deposits are similar in scale to those seen in the as-received NCG powder, Figure 9c.

Although pure NCG feedstock with an initial graphite fraction of 41% could be deposited as single tracks, it did not prove to be suitable for coating deposition. When attempting to deposit a coating from partially overlapping single tracks, each new track eroded the previously deposited track adjacent to it, so that no coating was formed. Coating deposition was only successful with Ni + NCG feedstock mixtures with graphite volume fractions of up to 30 vol%.

Tracks deposited onto aluminium retained a higher graphite fraction than those deposited onto titanium, reaching a maximum of 17.4 vol%. Given this finding, coatings of three different compositions were deposited onto aluminium for pin-on-disc testing. The same deposition conditions were used for each coating composition: traverse rate of 40 mms$^{-1}$, track overlap of 50%, and laser power 1500 W. The graphite content and porosity of the coatings produced are detailed in Table 4.

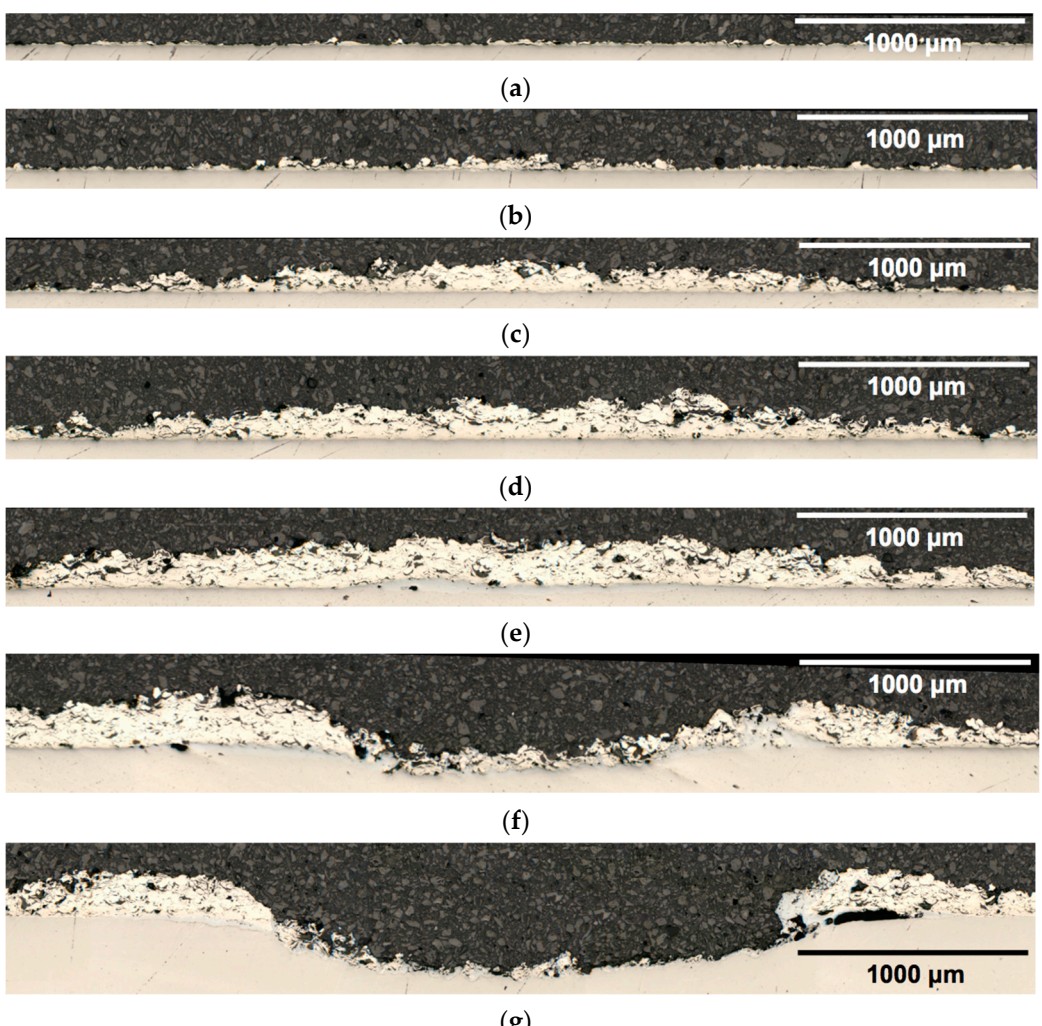

**Figure 6.** Deposition of mixtures of nickel and NCG at a total of 30 vol% graphite on titanium using laser powers ranging from 0–1744 W. (**a**) 30 vol% graphite mixture on Ti—0 W. (**b**) 30 vol% graphite mixture on Ti—99 W. (**c**) 30 vol% graphite mixture on Ti—492 W. (**d**) 30 vol% graphite mixture on Ti—805 W. (**e**) 30 vol% graphite mixture on Ti—1118 W—Substrate melting visible in centre. (**f**) 30 vol% graphite mixture on Ti—1432 W. (**g**) 30 vol% graphite mixture on Ti—1744 W.

**Table 4.** Graphite content and porosity in the coatings prepared for wear testing.

| Feedstock Composition (vol%) | Porosity in Single Track (vol %) | Porosity in Coating (vol%) | Graphite in Single Track (vol%) | Graphite in Coating (vol%) |
|---|---|---|---|---|
| Nickel | 1.1 | 0.9 | 0 | 0 |
| Ni/17 Graphite | 1.5 | 6.3 | 4.1 | 3.1 |
| Ni/30 Graphite | 6.3 | 2.5 | 17.4 | 13.3 |

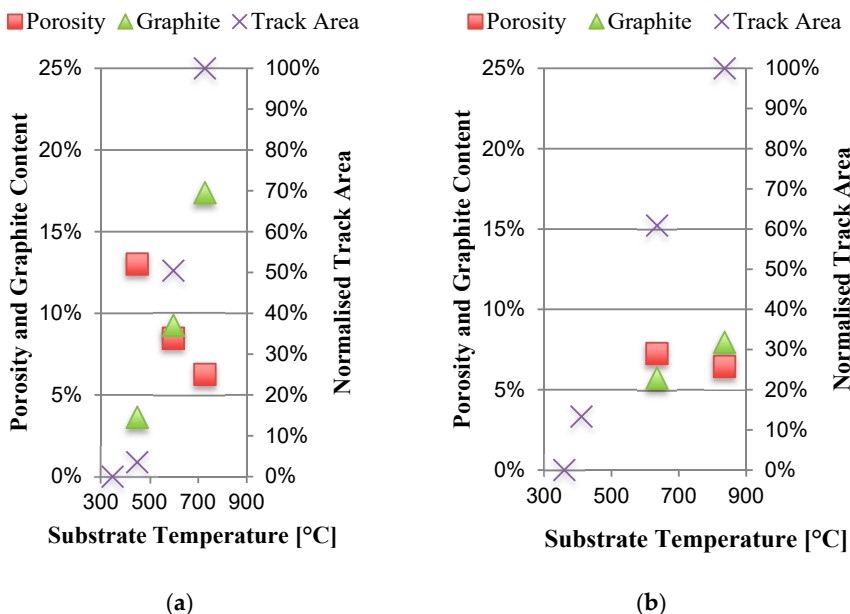

**Figure 7.** Porosity, graphite content, and cross-sectional area as a function of temperature on aluminium (**a**) and titanium (**b**) substrates for the Ni/NCG feedstock containing 30 vol% graphite at a traverse rate of 40 mms$^{-1}$.

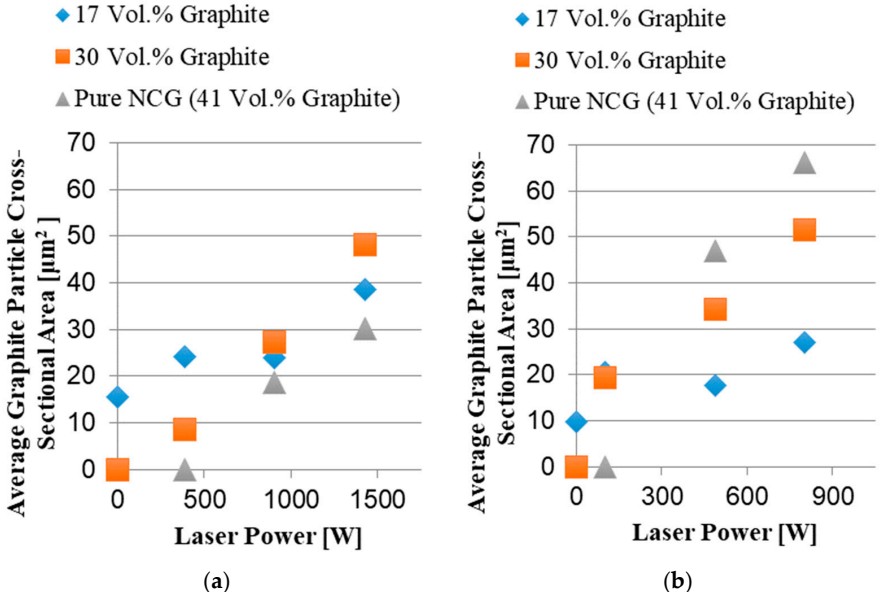

**Figure 8.** Graphite particle size for single lines deposited on aluminium (**a**) and titanium (**b**).

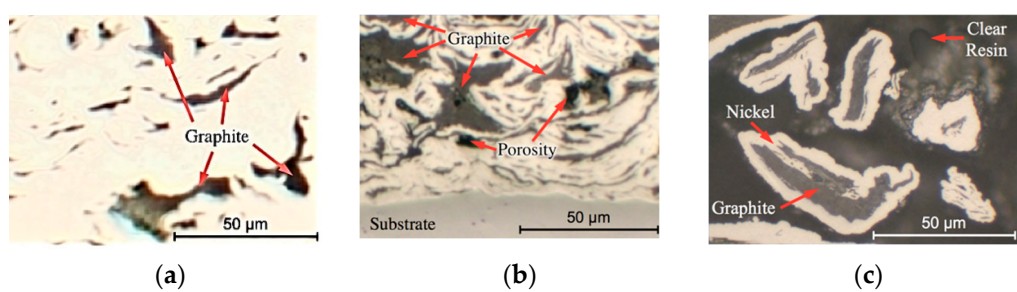

**Figure 9.** Micrographs showing (**a**) NCG particles co-deposited with nickel powder, (**b**) NCG alone, and (**c**) raw NCG powder before spraying for comparison.

### 3.2. Tribological Properties

Pin-on-disc testing took place using coatings containing 0, 3.1, and 13.3 vol% graphite. A minimum of three tests were carried out for each composition to assess consistency. Samples were tested without surface finishing to ensure contamination did not influence the results. Representative images of worn test samples are shown in Figure 10.

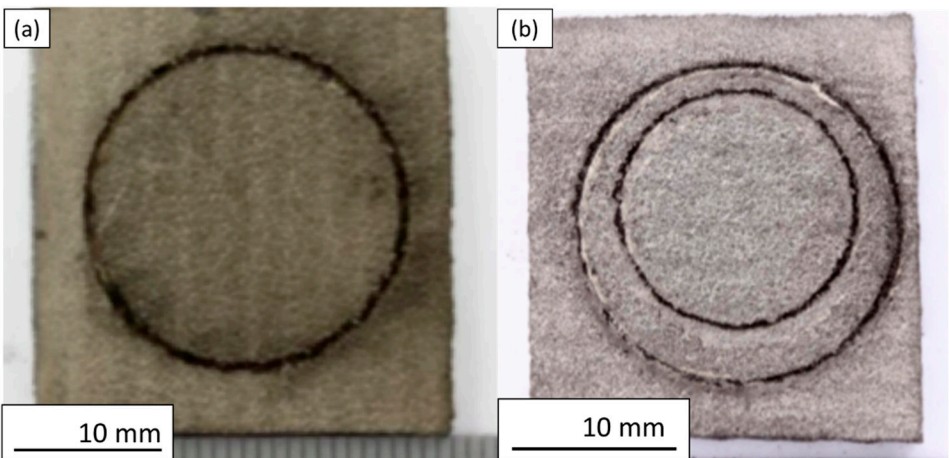

**Figure 10.** Wear scars from pin-on-disc testing of (**a**) nickel and (**b**) Ni + NCG 13.3 vol% graphite coating.

For each of the three graphite contents tested, CoF was consistent across at least three samples. All samples experienced the same abrasive wear mechanism, confirmed by post-inspection of all samples by ESR, as the speeds used are within the same order of magnitude.

Figure 11 shows a graph of representative friction data from coatings containing each graphite content tested. The CoF for the nickel coating rose rapidly to 0.5, before steadily climbing to 0.62 during the test. The Ni/NCG coatings both showed reduced friction. Only the 13.3 vol% coating maintained a stable CoF (average 0.135) throughout the test.

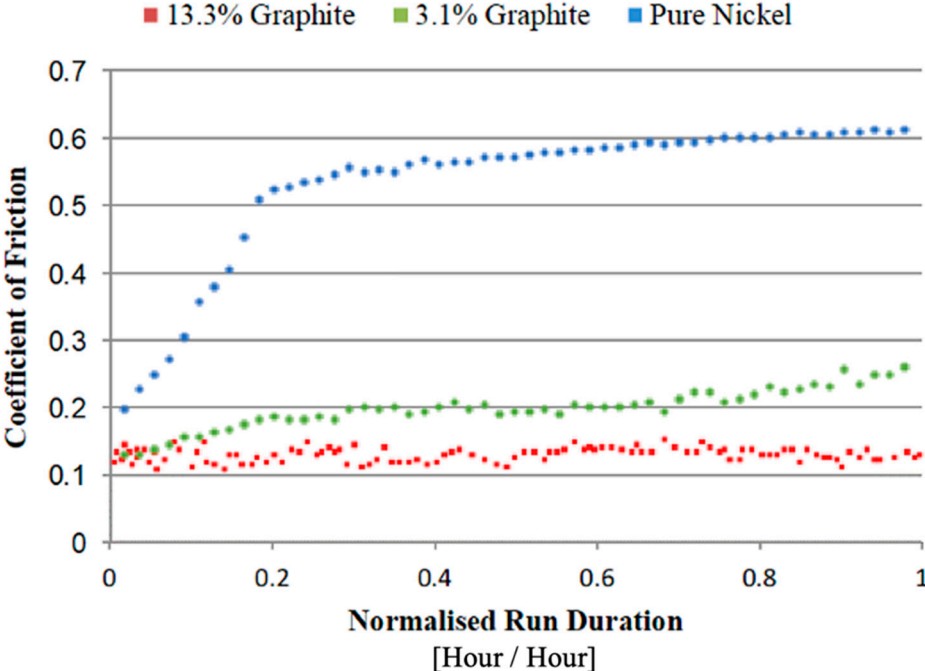

**Figure 11.** Representative data sets for the evolution of CoF versus time for the three coating compositions tested.

Wear rate was calculated from measurements of volume removed from the samples using white light interferometry; an example of a typical wear surface is shown in Figure 12.

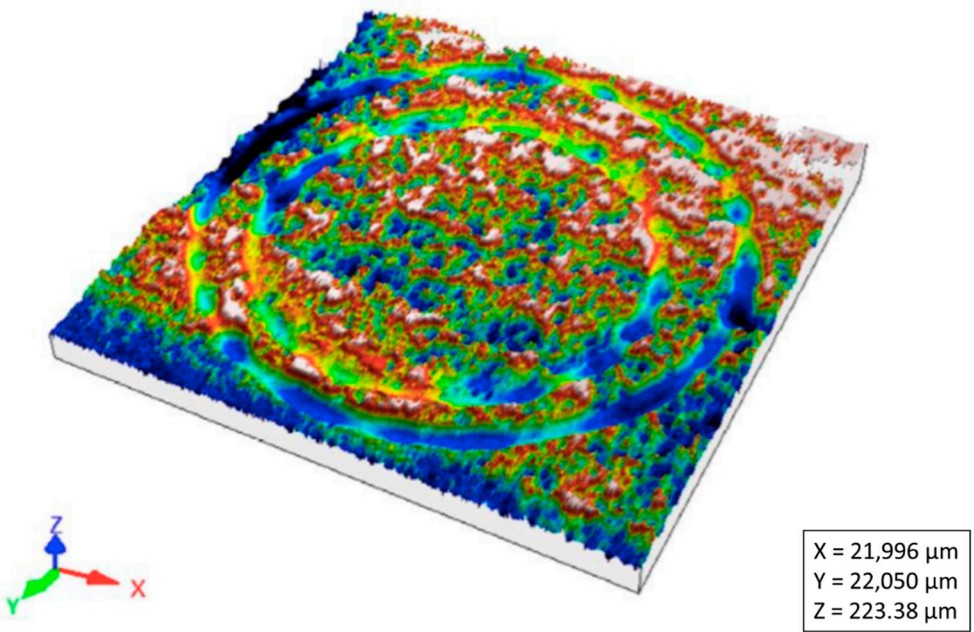

**Figure 12.** White light interferometer scan of wear scars on 13.3 vol% graphite coating.

CoF and wear rate are summarized for each composition in Figure 13. Each data point is an average of three tests with the displayed error bars showing the standard deviation for each measurement.

**Figure 13.** Friction and wear results plotted against graphite content. Error bars show the standard deviation for each measurement.

## 4. Discussion

### 4.1. Deposition and Graphite Inclusion

As Figure 4 shows, although nickel deposited using CS, a minimum laser power needed to be to be exceeded before deposition occurred for the feedstocks containing uncoated graphite. This behaviour is common in SLD and suggests that a minimum

deposition site temperature must be surpassed before deposition will occur, demonstrating that the presence of uncoated graphite particles inhibits bonding during CS deposition.

While feedstocks containing either coated or uncoated graphite did not match the build rate of nickel alone, when laser heating was employed, the 17 vol% graphite Ni/NCG powder mixture resulted in build rates close to twice those achieved using 17 vol% uncoated graphite. This indicates that the nickel shells on the NCG particles significantly aid deposition and bonding between the nickel powder and NCG. One point to note is that the cold spray build rate of the 17 vol% NCG feedstock was only 21% of that of pure nickel on the titanium substrates, whereas it was 64% of nickel on aluminium. This may be due to the greater hardness of the titanium substrates (grade 2 titanium has 50% greater hardness than Al6082 at room temperature [27,28]) leading to increased forces on the particles on impact, disrupting the nickel shells on the NCG particles to a greater extent, and leading to more exposed graphite, inhibiting bonding.

Cross sections of Ni, Ni + 17 vol% graphite, and Ni + NCG with 17 vol% graphite deposits are shown in Figure 5. The Ni + graphite deposit has only 56% of the cross-sectional area of the Ni + NCG coating deposited using the same conditions, and it contains no visible graphite flakes. It is unlikely to provide a significant friction reduction when compared with nickel alone.

Retained graphite and porosity for deposits on both aluminium and titanium prepared using a 30 vol% graphite Ni + NCG feedstock are shown in Figure 7. The maximum retained graphite fraction of 17 vol% was obtained using an indicated temperature of 728 °C (above the melting point of aluminium) on an aluminium substrate. Note that due to uncertainties regarding the appropriate value of emissivity to use while depositing nickel on aluminium, and the presence of laser-heated powder, this temperature is likely to be inaccurate. The primary purpose of the pyrometer in SLD is process control. As such, consistency rather than accuracy is prioritized for the temperature reading.

Graphite fractions observed on titanium substrates are lower than those on aluminium, reaching a maximum of only 8 vol% at an indicated deposition site temperature of 836 °C. For both aluminium and titanium substrates, the retained graphite fraction in the deposits is maximized at the onset of substrate erosion.

Polished sections of deposited tracks, depicted in Figure 9, show that although Ni + NCG did not deposit via cold spray; elevating the deposition site temperature via laser heating led to a deposit structure which is similar to that produced through cold spraying NCG alongside soft materials such as copper and aluminium, as reported in [17].

The trend in Figure 8 shows particle size increasing with laser power, up to the point at which substrate melting takes place. Substrate melting sets a limit on the size of the graphite particles incorporated into the deposits. The average measured cross-sectional graphite core size for as received NCG powder was 104 $\mu m^2$, whereas the largest average size obtained in Ni/NCG coatings was 66 $\mu m^2$ at a pyrometer temperature of approximately 800 °C on titanium.

Coatings deposited for pin-on-disc testing are summarized in Table 4. For each coating the graphite content reduced when going from single tracks to coatings, suggesting that the deposition of each successive pass led to the erosion of the graphite portion of the preceding track. Despite this, the 30 vol% graphite feedstock resulted in a coating with a lubricant content in the 6–20 vol% range for effective lubrication described in the literature.

### 4.2. Tribological Properties

The CoF displayed by the pure nickel coating rose beyond 0.6 during testing. This approaches the typical values found for nickel on steel (0.64) [29].

Both the coatings containing NCG showed significant reductions in friction, with the 13.3 vol% graphite coating displaying a consistent CoF of 0.135 ± 0.003, representing an 80% reduction in friction when compared with the pure nickel coating, and a similar reduction in friction compared to that reported for aluminium and steel (0.61) [29]. A CoF of 0.135 is in line with the values claimed for sintered bronze/graphite bearings [30], and is

lower and more stable than the values reported for Al/NCG deposited using CS [17]. 0.135 also approaches the 0.1 CoF of a graphite block on steel [29], confirming that the graphite in the deposited coating is offering effective lubrication.

The data for the 13.3 vol% coatings demonstrate that when sufficient amounts of graphite are incorporated into a Ni/NCG coating, the coefficient of friction is stable over time.

The measured wear rate is less consistent than the CoF for all three compositions. Although the pure nickel coatings displayed the lowest average wear rate, the uncertainties in the measurements overlap, making it difficult to be certain that the observed pattern represents the true wear behaviour. The average wear rate of the 13.3 vol% graphite coatings was $1.63 \times 10^{-13}$ Nm/m$^3$. This compares favourably with wear rate reported for 1050 series aluminium, $1.56 \times 10^{-12}$ Nm/m$^3$ [7], but does not represent an improvement over a pure nickel SLD coating.

## 5. Conclusions

1. SLD can deposit nickel/graphite coatings onto aluminium and titanium substrates when nickel shells are used to encapsulate the lubricant.
2. Uncoated graphite cannot be incorporated into nickel coatings using either CS or SLD.
3. Laser heating is required for the deposition of Ni/NCG coatings deposited using a feedstock containing 30 vol% graphite.
4. For a given graphite volume fraction in the feedstock, deposit dimensions, graphite vol%, and graphite particle size all increase with laser power and deposition temperature.
5. Ni + NCG feedstock containing 30 vol% graphite has been used to deposit Ni/13.3 vol% graphite coatings using SLD. This graphite fraction is in the range identified as providing effective lubrication in the literature.
6. The average wear rate of the 13.3 vol% graphite coatings, $1.63 \times 10^{-13}$ Nm/m$^3$, is comparable with SLD nickel and less than that reported for aluminium substrates.
7. Coatings containing 13.3 vol% graphite have a low and stable CoF, $0.135 \pm 0.003$. This improves upon the metal/solid lubricant coatings reported in the literature and is comparable to sintered composites produced via powder metallurgy.

**Supplementary Materials:** The following supporting information can be downloaded at: https://www.mdpi.com/article/10.3390/coatings12060760/s1, the excel files for the wear tests.

**Author Contributions:** Conceptualization, N.S. and M.S.; methodology, N.S. and A.C.; investigation, N.S.; writing—original draft preparation, N.S. and A.C.; writing—review and editing, A.C. and N.S.; supervision, M.S. and W.O.; funding acquisition, W.O. All authors have read and agreed to the published version of the manuscript.

**Funding:** This research was funded by Engineering and Physical Sciences Research Council, grant number EP/L016567/1.

**Informed Consent Statement:** Not applicable.

**Data Availability Statement:** The data presented in this study are available in supplementary information.

**Conflicts of Interest:** The authors declare no conflict of interest.

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
