# Peer review of "Deposition of Self-Lubricating Coatings via Supersonic Laser Deposition (SLD)"

_coatings, doi:10.3390/coatings12060760_

Round 1

Reviewer 1 Report

The information presented in this article is interesting, but the article needs a major revision. Comments are written according to the sections:

1. Abstract

The abstract must be redone. In this version of the abstract, half of the information is about the SLD method. There is little information that relates to the work. The abstract is a brief summary of your research, so make corrections accordingly.

2. Introduction

In the introduction the authors give information about the methods of application of composite materials, but from the presented information it is not quite clear why aluminum and titanium were chosen as substrates. I suggest adding information about these materials in the Introduction so that readers can understand the logic of choosing these substrates.

3. Materials and Methods

- Section 2.1. SLD Equipment. In this section, you should change the sequence of the material: starting with the text, then figure is placed. Please correct it.

- Section 2.2. Materials. It is also necessary to change the sequence in which the data appear. I advise you to do the following. After the sentences «Details of the feedstock powders are provided in Table 1. Micrographs showing polished cross sections of the three powder types are shown in Figure 2.» locate the Table 1 и Figure 2. After that, provide the text «Powder mixtures were shaken together in sealed containers for 5 minutes prior to being loaded into the powder feeder in order to ensure powder mixing. The feedstock was homogeneous and flowed evenly when loaded into the feeder. All powders were stored in a heated cabinet to prevent moisture condensation. Similarly, a heating blanket strapped around the powder feeder was used to keep the feedstock powder between 50°C and 55°C once loaded in the feeder. Throughout this paper, all constituents in a powder mixture are reported by their volume fraction. An SEM image of a mixture of nickel and NCG feedstock powders help illustrate the difference in particle size and shape, a mixture of two different powders, NCG (17 % graphite by volume) and pure nickel, is shown in Figure 3.» After that locate the Figure 3.

- Section 2.3. Deposition. Change «Table 1: Parameters for single line deposition trials» on «Table 2. Parameters for single line deposition trials».

- Change section 2.3. Characterisation on Part 2.4. Characterisation. In the same section change «The specific wear testing regime and settings used are listed in Table 2.» on «The specific wear testing regime and settings used are listed in Table 3.» and also change «Table 2: Load parameters for the wear test» on «Table 3. Load parameters for the wear test.»

4. Section 3. Results & Discussion

Authors should be divided into two separate sections: 3. Results, 4. Discussion. Please take these comments into consideration and add the necessary information to Discussion.

5. Figures

Check the numbering of the figures! Figures 5, 6, 7, 8, 9, 10 and 11 should appear after figure 4. Now the numbering is wrong, please correct it. Don't forget to change the text accordingly.

6. References

It is necessary to increase the number of literature sources used. It is not serious to use only 14 references for the article. Increase their number to 30.

7. There are some typos in the test, please check and correct them.

Author Response

Dear Reviewer 1,

Thankyou for your comments.

  1. We have altered the abstract
  2. We have added information regarding our choice of substrate and lubricant material.
  3. We have altered the order in which text and figures appear.
  4. We have separated the result and discussion sections.
  5. We could not replicate the problems you found with labelling but have reinserted the labels and cross references in the hope of fixing this issue.
  6. We have updated our literature search to cover more recent publications.
  7. We have proof read the text to remove typos where we could find them.

Regards

Andrew Cockburn

Reviewer 2 Report

Nicholas Soane et al., investigate the properties of Ni-Graphite composite coatings on Titanium and Aluminum as a function of powder type, feedstock composition and various process conditions. The Ni-graphite composite coating with 13.3 vol% of graphite achieved a co-efficient of friction less than 0.14 showing improved lubrication properties.

The authors need to rewrite the abstract to reflect the key findings more prominently. At present the findings are buried under general information of the topic. For example, some of the key findings in conclusion can be incorporated into the abstract.

From the table it looks like the Graphite volume percentage in the feed stock has been varied from 0-41%. However, only selected results have been shown like 17% and 30%. Why were only these results selected?

Author Response

Dear reviewer 2

Thankyou for taking the time to review our paper.

  1. We have re written the abstract to reflect your comments.
  2. We have highlighted why we could not carry out wear testing with a coating made from 41vol% graphite.

Regards

Andrew Cockburn

Reviewer 3 Report

This article presented with quality the study of the use of SLD to deposit nickel/graphite composite coatings on aluminum and titanium substrates. Their results were satisfactory and the article should be edited to improve its quality.

  1. In the introduction, the information cited in lines 45 and 63 needs a reference.
  2. The figures in the results were numbered incorrectly. As for example in line 147, 150, 153, 158, 167.
  3. The conclusions need to be improved. For example, better clarify the results of tribology properties.

Author Response

  1. Citations have been added.
  2. Figure numbering has been addressed.
  3. Conclusions have been modified to clarify the tribological results.

Reviewer 4 Report

The article needs the following improvements:

  • the introduction does not allow the assessment of the current knowledge in the field of deposition of self-lubricating coatings via supersonic laser deposition because the cited literature covers the last 5 years only in 3 cases, the remaining ones are publications from before 2016. The introduction should be supplemented with the latest research results
  • the purpose of the research should be clearly stated; explain why such materials were chosen
  • all abbreviations should be defined the first time they appear in the text
  • the numbering of figures should be checked and corrected (also in the text). Figures are repeatedly numbered.
  • detailed comments have been made directly in the PDF file

Author Response

Dear Reviewer 4.

Thankyou for your comments.

  1. We have updated our literature review to include more recent work in this area.
  2. We have clarified he purpose of this work in the introduction.
  3. The rational behind the choice of materials is provided in the penultimate paragraph in the introduction (lines 101-109) 
  4. Figure numbering has now been addressed.
  5. The edits highlighted in the pdf have been carried out.

Reviewer 5 Report

The article tests the use of SLD to 16 deposit nickel/graphite composite coatings on aluminum and titanium substrates. The topic of the article is interesting for the selected journal, however some points have to be addressed in order to improve the manuscript and to make it flawless as possible. This is why the article is suggested for major revision. Authors are suggested to improve following issues:

General remarks:

Please correct all the typos (example: MoS2 etc.)

Please correct the units and style of writing them to one selected style.

Figure 2: please mark the features on the figures.

Figure 3: what does represent blue line under the images, scale?

Please add to each machine/microscope used in the script, also the company, city and country of the origin, example Olympus BX51, Olympus Europa, Hamburg, Germany.

On page 4 please correct spacing between the sentence and the Table 2.

Table 2: under run time replace minutes with min.

Figure 4: please change the color of 17% Graphite, because it is hard to distinguish between the data and background.

Please correct the numbering of figures and correct them in the text accordingly.

Figure 5: Please correct it, so it isn´t so blurry.

Figure 9 a) Please replace the micrograph with another one, because the resolution of the image is not appropriate.

Figure 10 (tribology): if there are at least 3 measurements, where is the standard deviation of measurements?

Specific remarks:

  1. Line 33: authors state that proportion of lubricant is dependent on load, matrix and type of lubricant. What about the number, size and volume fraction of precipitates? Please explain and add to introduction.
  2. Updated the results and discussion part in correlation to precipitates effect on wear resistance and friction.
  3. Chapter 3.2.: -Please add the images of tested samples with pin-on-disc and mark the wear mechanism.

-Please correct the graphs to include standard deviation.

-Line 210: please add the reference (literature value).

-Please provide in % the improvement of wear and friction resistance compare to each coating.

  1. Update conclusions accordingly and it is suggested to formulate them with numbering.

Author Response

Dear Reviewer 5,

Thankyou for your comments regarding the paper.

  1. The Typos and formatting inconsistencies have been addressed.
  2. Features have been labelled in figure 2.
  3. The line under figure 3 has been removed.
  4. Greater detail has been added to the description of the origin of the equipment used.
  5. Table 2 formatting has been revised.
  6. minutes has been replaced by min in table 3.
  7. A higher resolution version of figure 5 has been prepared.
  8. A higher resolution version of 9(a) has been inserted.
  9. The standard deviations for the CoF and wear rate are now displayed as error bars in figure 13.
  10. We have modified the introduction to clarify our statement regarding lubricant volume.
  11. We are unclear about the comment relating to precipitates. There are no precipitates in the coatings, only added graphite.
  12. Images of worn test samples have been added.
  13. The % improvement in CoF is stated. The standard deviation in measured wear rates is large, precluding expressing change in wear rate as a percentage.
  14. Regarding figure 4. The 17 vol % graphite as NCG is shown as orange squares on white. The 17 vol% uncoated graphite data is shown as green crosses on white. We do not understand why these would be hard to see. Can the reviewer clarify this?
  15. Regarding line 210, a reference has been provided from a bearing  supplier.
  16. The % improvement in CoF has been added. The wear rate has a large standard deviation making the provision of a percentage change in average wear rate potentially misleading. The wear rate did not show an improvement when going from pure Ni to a Ni/NCG sld coating.
  17. Conclusions have been updated and numbered.

Round 2

Reviewer 1 Report

Colleagues, thank you for your work! I can see that you have made significant changes to the Manuscript. However, small corrections need to be made before this study can be published: 

1. Please check the font size used in the tables. In Tables 4 and 3 use a smaller font than in Tables 1 and 2. Please make everything look the same. Perhaps you should read the requirements of the template again.

2. Clarification for Table 1: should it be that Nominal Particle Sizes of Nickel Coated Graphite (NCG) are -90 +30 µm? Please check the correctness of this statement, the negative particle size is confusing.

3. Lines 169, 170: "Rockwell hardness Cwas". Clearly missing a space. Please change Cwas on C was. 

4. Lines 178, 179: "used are listed in Table 3". Please change Table 3 on Table 3. Also Table 3 should be on Line 178, please correct.

5. To the caption of Figure 4 should apply a text alignment to the width.

6.  Figure 10. Please make the same size for Figures (a), (b).

7. Figure 13. I suggest to change the value for coefficient of Friction  0.000 ÷ 0.700 on 0.0 ÷ 0.7.

8. Lines 324, 325: "......is comparable with sintered bronze/graphite bearings[30] 135 is approaching the....". Change bearings[30] on bearings [30]. It is not clear which value is comparable with sintered bronze/graphite. Check if the value of 135 is correct. 

9. Please check the Manuscript again for any typographical errors, mistakes or merged words.

Author Response

Dear Reviewer 1

Thankyou for your comments.

1). We have ensured that the fonts used in the tables are consistent

2). We have altered the way the size data is presented in table 1.

3). A space has been added after Rockwell hardness C.

4). We have corrected the cross reference to table 3.

5). We have corrected the width of table 4.

6). We have resized figure 10 (b) and added scale bars to 10 (a) and 10 (b).

7). We have modified the scale on the y axis of figure 13.

8). The CoF of 0.135 is comparable with the values given by the bearing supplier listed in reference 30.

9). We have re-read the test for typos.

Reviewer 4 Report

thank you for the answers, I find them satisfactory

Author Response

Dear Reviewer 4,

Thank you for your comments.

Regards

Andrew Cockburn

Reviewer 5 Report

The authors of the article "Deposition of Self-Lubricating Coatings via Supersonic Laser Deposition (SLD) ", have significantly improved the manuscript and have incorporated all of the suggestions/recommendations. In addition, the graphical presentation of results and testing goes now nicely hand by hand with the text. The topic is now appropriately developed from the introduction to the conclusion. The topic is of high interest for the publication in Coatings due to its originality and advances in methods and observations. On these grounds, I believe that the manuscript should be accepted in the present form.

Author Response

Dear Reviewer 5,

Thank you for your comments

Regards

Andrew Cockburn